# Effect of Mechanical Heterogeneity on the Structural Integrity of HTPB Propellant Grain

**DOI:** 10.3390/ma16134590

**Published:** 2023-06-25

**Authors:** Xiangyang Liu, Buqing Hui, Hui Wang, Hang Chen, Dongmo Zhou

**Affiliations:** 1School of Mechatronic Engineering, North University of China, Taiyuan 030051, China; liuxy@bit.edu.cn (X.L.); huibuqing@163.com (B.H.); 18234163863@163.com (H.W.); 18161012381@163.com (H.C.); 2School of Aerospace Engineering, Beijing Institute of Technology, Beijing 100081, China

**Keywords:** mechanical heterogeneity, HTPB propellant, gradient finite element, structural integrity, safety factor

## Abstract

To investigate the structural effects of the mechanical heterogeneity of Hydroxyl-terminated polybutadiene (HTPB) propellant grain under ignition pressurization, a gradient finite element method was proposed to evaluate its structural integrity. The heterogeneous mechanical properties of the propellant grain were constructed and assessed. The results demonstrate that the mechanical properties of the propellant grain are spatially variable when taking into account the effect of the load. The range of variation in the mechanical properties is related to the size of the load and its effect on the mechanical properties of the propellant. Two key parameters that affect the mechanical response of the grain are the non-uniform distribution of the modulus and the damage strain threshold. An increase in the propellant modulus leads to an increase in the stress response and a decrease in the strain response of the propellant grain under ignition pressurization. Meanwhile, an increase in the damage strain threshold improves the propellant’s modulus in the linear elastic stage in a disguised form. This also leads to an increase in the stress response and a decrease in the strain response when the strain response exceeds the damage strain threshold. The safety factor, based on the equivalent strain failure criterion of the grain, directly depends on both the strain response of the propellant grain and the maximum elongation of the propellant. Furthermore, the change in the safety factor of two propellant grains is primarily affected by the maximum elongation of the propellant.

## 1. Introduction

The propellant grain is a crucial load-bearing component of a solid rocket motor (SRM). Its structural integrity determines the reliable operation and storage life of SRMs [1,2]. Throughout the life cycle of an SRM, the propellant grain must withstand various types of loads, including temperature, gravity, vibration, ignition impact, and axial overload, among others [3,4,5]. These loads can cause complex physical and chemical changes in the internal structure of the propellant grain over time [6], ultimately leading to an uneven distribution of the mechanical properties of the propellant grain. In other words, different parts of the propellant grain exhibit different mechanical properties. The traditional structural integrity assessment method, which assumes that the propellant’s mechanical properties are uniform, cannot accurately reflect these differences. Therefore, it is essential to evaluate the structural integrity of the propellant grain while considering its mechanical heterogeneity to improve the accuracy of the assessment.

When considering the uneven distribution of mechanical properties in the propellant grain, the mechanical property parameters become functions of space coordinates. Since the basic equations of mechanics are usually variable coefficients, it is difficult to obtain an analytical solution mathematically, and the most popular approach is the finite element method (FEM) [7,8]. However, the main issue encountered when using FEM for mechanically heterogeneous materials concerns modeling continuously varying material properties [9]. The simplest way to model a mechanically heterogeneous material involves using conventional homogeneous elements in successive layers of the mesh, each with its own material properties. This approach is known as the layered finite element method [10,11,12,13]. The main drawback of this method is the non-continuous segmented distribution of material properties, which requires a fine mesh to achieve accuracy, leading to excessive computational costs. Moreover, the preparation of input data and adjusting the mesh of different gradation regions is quite cumbersome in this approach as well [9]. To avoid these drawbacks, it is important to include the gradient of material properties in the model at the element level to retain accuracy when using coarser meshes. Therefore, an isoparametric finite element has been developed and applied to solve problems involving mechanically heterogeneous materials [14,15,16,17]. However, this approach still requires interpolating material properties at each Gaussian integration point from the nodal material properties of the element using isoparametric shape functions, which need to construct the element stiffness matrices related to coordinates. For complex structural components of heterogeneous materials, this approach has problems with the complicated construction of element stiffness matrices and heavy workload. Other methods for dealing with material heterogeneity, such as mesh-free methods [18,19], microelement methods [20,21,22], and extended multi-scale finite element methods [23,24,25], are still computationally expensive and difficult to apply to complex models.

Therefore, the gradient finite element method was used to evaluate the effect of the mechanical heterogeneity of propellant mechanical performance on the grain structural integrity. In a case-bonded SRM, the propellant grain undergoes various constant strains introduced during curing and cooling, resulting in heterogeneous mechanical properties. Therefore, the mechanical heterogeneity of propellant grain is constructed by correlating the spatial distribution of constant strains, and the effect of mechanical heterogeneity on the structural integrity of the propellant grain is investigated. Figure 1 presents the organization of this paper.

## 2. Analysis and Modelling

### 2.1. Structural Integrity Assessment Method

The essence of the gradient finite element method is to analyze the structural integrity of the propellant grain by specifying the material property variation function. This method can effectively examine the effects of the mechanical heterogeneity of the propellant on the grain’s structural integrity, and the number of units, degrees of freedom, and computational workload of this method are lower. It is highly adaptable to complex geometries and various physical models.

The process for evaluating the structural integrity of the propellant grain using the gradient finite element method is as follows: Firstly, establish the relationship between the loads (such as temperature, stress, strain, etc.) and the mechanical constitutive properties of the propellant and determine the basis for the mechanical heterogeneity of HTPB propellant. Secondly, one must analyze the load field of the propellant grain and examine load parameters at each integration point. Then, the constitutive parameters must be accurately assigned to the integration points using the ABAQUS UMAT subroutines, and the spatial distribution field of the mechanical properties in propellant grain needs to be established. Finally, we analyze the mechanical responses of the propellant grain under specific working conditions and assess the structural integrity according to the propellant failure criterion.

### 2.2. Constitutive Model

#### 2.2.1. Stress-Strain Curve

Previous studies show that there are two typical characteristics of HTPB propellant constitutive characteristics under constant strain aging. 

Type I: The constant strain $\varepsilon_{0}$ has no effect on the stress-strain response in the linear elastic stage, and there is an overlap of stress-strain curves between different constant strains, but it decreased the damage strain threshold $\varepsilon_{th}$ and stress response after the onset of damage in the damage stage. The constants can increase the maximum elongation $\varepsilon_{m}$ of the HTPB propellant, but have almost no effect on the modulus in the linear elastic stage ($E_{e}$) and in the damage stage ($E_{d}$), which are approximately equal to the slope of each stage, as shown in Figure 2a. The details refer to Ref. [26].

Type II: The constant strain $\varepsilon_{0}$ can decrease the modulus, damage strain threshold $\varepsilon_{th}$, and maximum elongation $\varepsilon_{th}$ of the HTPB propellant. Referring to the test data in Ref. [27], the stress-strain curves of the HTPB propellant under different constant strains are constructed, as shown in Figure 2b.

#### 2.2.2. Constitutive Model

The viscoelastic damage constitutive models adopted in this paper are provided in Ref. [26], which are presented as follows:(1)$$\sigma =(1-q\cdot D)[f_{e}(\varepsilon )+E_{1}\int_{0}^{t}\dot{\varepsilon }\exp(\frac{t-\tau }{\theta })]d\tau$$
(2)$$f_{e}(\varepsilon )=\sigma_{m}[1-\exp(-\alpha \varepsilon )]$$
(3)$$D=\left\{\begin{array}{ll} 0 & \varepsilon \leq \varepsilon_{th}\\ 1-\exp[-{\left(\frac{\varepsilon -\varepsilon_{th}}{\eta }\right)}^{m}] & \varepsilon \geq \varepsilon_{th} \end{array}\right.$$
where q is the initial damage coefficient and the value range is 0–1, D is the damage variable, $f_{e}(\varepsilon )$ is the nonlinear term of the constitutive model, $E_{1}$ and θ are the elastic constants and relaxation time of the material, $\dot{\varepsilon }$ is the strain rate, $\sigma_{m}$ and α are material parameters, $\sigma_{m}$ represents the limit value of $f_{e}(\varepsilon )$ when the strain approaches infinity, α is the ratio of initial modulus to $\sigma_{m}$ under equilibrium, $\varepsilon_{th}$ is the damage strain threshold, and η and m are scale parameters and shape parameters of the Weibull distribution function.

The constitutive model parameters of two types of propellant are shown in Table 1.

According to the results of Refs. [26,27], the relationship between the maximum elongation $\varepsilon_{m}$ of type I and type II propellants and the constant strain in Figure 2 is shown in Table 2.

#### 2.2.3. Increment Alization of the Constitutive Equations

Direct implementation of the developed constitutive equations into a finite element program is not feasible. For any constitutive equation used in the finite element formulation, it is necessary to establish the relationship between the increase in strain (or deformation rate) and the increase in stress. In order to describe the structural responses of a propellant grain during the ignition process, the incremental constitutive relation was adopted and illustrated as follows.

The equation is based on decomposing the stress and strain within an isotropic body into deviatoric (shear) and hydrostatic (volumetric) components as [28,29,30]:(4)$$\sigma_{ij}=S_{ij}+\frac{1}{3}\delta_{ij}\sigma_{kk}\;\sigma_{kk}=\sigma_{11}+\sigma_{22}+\sigma_{33}$$
(5)$$\varepsilon_{ij}=e_{ij}+\frac{1}{3}\delta_{ij}\varepsilon_{kk}\;\varepsilon_{kk}=\varepsilon_{11}+\varepsilon_{22}+\varepsilon_{33}$$
where $\delta_{ij}$ is the Kronecker delta (1 for *i* = *j* and 0 for *i* ≠ *j*), $S_{ij}$ and $e_{ij}$ are the stress deviatoric tensor and strain deviatoric tensor, respectively. $\sigma_{kk}$ and $\varepsilon_{kk}$ are the volumetric stress and volumetric strain, respectively. $\sigma_{ii}$ and $\varepsilon_{ii}$ (*i* = 1, 2, 3) are the normal stress and normal strain, respectively, where
(6)$$S_{ij}=2Ge_{ij}$$
(7)$$\sigma_{kk}=3K\varepsilon_{kk}$$
where G=E/2(1+v) and K=E/3(1−2v) are the shear modulus and volumetric modulus, respectively. v is Poisson’s ratio.

When damage is not considered, the constitutive model without damage can be divided into the nonlinear term and the viscoelastic integral term in Equation (1). Assuming the propellant material is isotropic, the elastic modulus of the nonlinear term can be obtained by the phenomenological method:(8)$$E(\varepsilon_{v})=\frac{\partial \sigma_{v}}{\partial \varepsilon_{v}}=\sigma_{m}\alpha \exp(-\alpha \varepsilon_{v})$$
where $\sigma_{v}$ and $\varepsilon_{v}$ are the equivalent stress and equivalent strain, respectively. Substituting Equation (8) into Equations (4), (6) and (7) yields the three-dimensional form of the nonlinear term.
(9)$$\sigma_{ij}^{Non}=\frac{E(\varepsilon_{v})}{1+v}e_{ij}^{Non}+\delta_{ij}\frac{E(\varepsilon_{v})}{3(1-2v)}\varepsilon_{kk}^{Non}$$
where $\sigma_{ij}^{Non}$ is the non-linear part of the three-dimensional stress, $\frac{E(\varepsilon_{v})}{1+v}$ is the shear modulus of the non-linear part, and $\frac{E(\varepsilon_{v})}{3(1-2v)}$ is the volumetric modulus of the non-linear part.

Similarly, the three-dimensional form of the viscoelastic term in Equation (1) is:(10)$$\begin{array}{c} \sigma_{ij}^{Visco}=2G\int_{0}^{t}\frac{\partial e_{ij}^{Visco}}{\partial \tau }\exp(-\frac{t-\tau }{\theta })d\tau \\ \;+\delta_{ij}K\int_{0}^{t}\frac{\partial \varepsilon_{kk}^{Visco}}{\partial \tau }\exp(-\frac{t-\tau }{\theta })d\tau \end{array}$$
where $\sigma_{ij}^{Non}$ is the viscoelastic part of the three-dimensional stress. $2G\int_{0}^{t}\frac{\partial e_{ij}^{Visco}}{\partial \tau }\exp(-\frac{t-\tau }{\theta })d\tau$ is the deviatoric stress of the viscoelastic part, and $K\int_{0}^{t}\frac{\partial \varepsilon_{kk}^{Visco}}{\partial \tau }\exp(-\frac{t-\tau }{\theta })d\tau$ is the volumetric stress of the viscoelastic part.

Assuming that the damage variable is isotropic, the damage variable is a function of the equivalent strain. The damage variable can be expressed as:(11)$$D(\varepsilon_{v})=1-\exp[-{\left(\frac{\varepsilon_{v}-\varepsilon_{th}}{\eta }\right)}^{m}]$$

Combining Equations (9)–(11), the three-dimensional form of the nonlinear viscoelastic constitutive model with damage is:(12)$$\begin{array}{ll} \sigma_{ij,D} & =[1-qD(\varepsilon_{v})]\times [\sigma_{ij}^{Non}+\sigma_{ij}^{Visco}]\\ & =[1-qD(\varepsilon_{v})]\times [\frac{E(\varepsilon )}{1+v}e_{ij}^{Non}+\delta_{ij}\frac{E(\varepsilon )}{3(1-2v)}\varepsilon_{v}^{Non}\\ & +2G\int_{0}^{t}\frac{\partial e_{ij}^{Visco}}{\partial \tau }\exp(-\frac{t-\tau }{\theta })d\tau \\ & +\delta_{ij}K\int_{0}^{t}\frac{\partial \varepsilon_{v}^{Visco}}{\partial \tau }\exp(-\frac{t-\tau }{\theta })d\tau ] \end{array}$$

The incremental form of the constitutive model is based on the above derivation. From Equation (9), the incremental form of the nonlinear part can be obtained:(13)$$\begin{array}{ll} \Delta \sigma_{ij,Non}^{t+\Delta t} & =\sigma_{ij,Non}^{t+\Delta t}-\sigma_{ij,Non}^{t}\\ & =\frac{E(\varepsilon_{v})}{1+v}\Delta \varepsilon_{ij,Non}^{t+\Delta t}+\delta_{ij}E(\varepsilon_{v})\lambda \Delta \varepsilon_{v,Non}^{t+\Delta t} \end{array}$$
where λ=v/[(1+v)(1−2v)].

The increment at time t+Δt is obtained by dividing the viscoelastic integral term into shear stress and volumetric stress. The shear stress increment of the viscoelastic integral term at time t+Δt is:(14)$$\begin{array}{ll} \Delta S_{ij}^{t+\Delta t} & =S_{ij}^{t+\Delta t}-S_{ij}^{t}\\ & =2G\int_{0}^{t+\Delta t}\frac{\partial e_{ij,Visco}^{t+\Delta t}}{\partial \tau }\exp(-\frac{t+\Delta t-\tau }{\theta })d\tau \\ & -2G\int_{0}^{t}\frac{\partial e_{ij,Visco}^{t}}{\partial \tau }\exp(-\frac{t-\tau }{\theta })d\tau \end{array}$$
where
(15)$$F_{ij}^{t}=\int_{0}^{t}\frac{\partial e_{ij,Visco}^{t}}{\partial \tau }\exp(-\frac{t-\tau }{\theta })d\tau$$

Assuming that $\dot{\varepsilon }$ is a constant value at time t+Δt, replaced by the average strain rate, Equation (14) can be expressed as:(16)$$\begin{array}{ll} \Delta S_{ij}^{t+\Delta t} & =2GF_{ij}^{t}[\exp(-\frac{\Delta t}{\theta })-1]\\ & +2G\theta \frac{\Delta e_{ij,Visco}^{t+\Delta t}}{\Delta t}[1-\exp(-\frac{\Delta t}{\theta })] \end{array}$$

Similarly, the volumetric stress increment is:(17)$$\begin{array}{ll} \Delta \sigma_{v}^{t+\Delta t} & =3KX_{v}^{t}[\exp(-\frac{\Delta t}{\theta })-1]\\ & +3K\theta \frac{\Delta \varepsilon_{v,Visco}^{t+\Delta t}}{\Delta t}[1-\exp(-\frac{\Delta t}{\theta })] \end{array}$$
where $X_{kk}^{t}=\int_{0}^{t}\frac{\partial \varepsilon_{kk,Visco}^{t}}{\partial \tau }\exp(-\frac{t-\tau }{\theta })d\tau$.

Substituting Equation (16) into Equation (17) yields the stress increment form of the viscoelastic integral term:(18)$$\begin{array}{ll} \Delta \sigma_{ij,Visco}^{t+\Delta t}=\Delta S_{ij}^{t+\Delta t}+\frac{1}{3}\delta_{ij}\Delta \sigma_{v}^{t+\Delta t} & =2GF_{ij}^{t}[\exp(-\frac{\Delta t}{\theta })-1]\\ & +2G\theta \frac{\Delta e_{ij,Visco}^{t+\Delta t}}{\Delta t}[1-\exp(-\frac{\Delta t}{\theta })]\\ & +\frac{1}{3}\delta_{ij}\{3KX_{v}^{t}[\exp(-\frac{\Delta t}{\theta })-1]\\ & +3K\theta \frac{\Delta \varepsilon_{v,Visco}^{t+\Delta t}}{\Delta t}[1-\exp(-\frac{\Delta t}{\theta })]\} \end{array}$$

Combining Equations (13) and (18), the increment form of the nonlinear viscoelastic constitutive model without damage can be expressed as: (19)$$\begin{array}{ll} \Delta \sigma_{ij}^{t+\Delta t} & =\Delta \sigma_{ij,Non}^{t+\Delta t}+\Delta \sigma_{ij,Visco}^{t+\Delta t}\\ & =\frac{E(\varepsilon )}{1+v}\Delta \varepsilon_{ij,Non}^{t+\Delta t}+\delta_{ij}E(\varepsilon )\lambda \Delta \varepsilon_{v,Non}^{t+\Delta t}\\ & +2GF_{ij}^{t}[\exp(-\frac{\Delta t}{\theta })-1]\\ & +2G\theta \frac{\left(\Delta \varepsilon_{ij,Visco}^{t+\Delta t}-\frac{\Delta \varepsilon_{v,Visco}^{t+\Delta t}}{3}\right)}{\Delta t}[1-\exp(-\frac{\Delta t}{\theta })]\\ & +\delta_{ij}KX_{v}^{t}[\exp(-\frac{\Delta t}{\theta })-1]\\ & +K\theta \frac{\Delta \varepsilon_{v,Visco}^{t+\Delta t}}{\Delta t}[1-\exp(-\frac{\Delta t}{\theta })] \end{array}$$

The dynamic damage evolution rate is expressed as the derivative of the damage variable with respect to time: (20)$$\frac{dD(\varepsilon_{v})}{dt}={\dot{\varepsilon }}_{v}\frac{m}{\eta }{\left(\frac{\varepsilon_{v}-\varepsilon_{th}}{\eta }\right)}^{m-1}\exp[-{\left(\frac{\varepsilon_{v}-\varepsilon_{th}}{\eta }\right)}^{m}]$$

When the time increment tends to infinity, $dD(\varepsilon_{v})$ can be approximated as $\Delta D(\varepsilon_{v})$, so the incremental form of the damage variable at time t+Δt is: (21)$$\Delta D(\varepsilon_{v})=\Delta {\dot{\varepsilon }}_{v}\frac{m}{\eta }{\left(\frac{\varepsilon_{v}^{t+\Delta t}-\varepsilon_{th}}{\eta }\right)}^{m-1}\exp\left[-{\left(\frac{\varepsilon_{v}^{t+\Delta t}-\varepsilon_{th}}{\eta }\right)}^{m}\right]$$

Combining Equations (19) and (21) can yield the incremental form of the nonlinear viscoelastic constitutive model with damage at time t+Δt: (22)$$\begin{array}{ll} \Delta \sigma_{ij,D}^{t+\Delta t} & =\sigma_{ij,D}^{t+\Delta t}-\sigma_{ij,D}^{t}\\ & =\left[1-qD(\varepsilon_{v}^{t+\Delta t})\right]\sigma_{ij}^{t+\Delta t}-\left[1-qD(\varepsilon_{v}^{t})\right]\sigma_{ij}^{t}\\ & =\left[1-qD(\varepsilon_{v}^{t+\Delta t})\right]\Delta \sigma_{ij}^{t+\Delta t}-q\Delta D(\varepsilon_{v}^{t+\Delta t})\sigma_{ij}^{t} \end{array}$$

The tangent modulus matrix can be expressed as the deviatoric derivative of the stress increment to the strain increment. When considering material damage, the damage variable takes the form of a scalar, so the tangent modulus with damage is:(23)$$C_{ijkl}^{t+\Delta t}=[1-D(\varepsilon_{v}^{t+\Delta t})]\frac{\partial \Delta \sigma_{ij}^{t+\Delta t}}{\partial \Delta \varepsilon_{kl}^{t+\Delta t}}-\sigma_{ij}^{t+\Delta t}\frac{\partial \Delta D(\varepsilon_{v}^{t+\Delta t})}{\partial \Delta \varepsilon_{kl}^{t+\Delta t}}$$
where
(24)$$\begin{array}{ll} \Gamma =\frac{\Delta \sigma_{ii}^{t+\Delta t}}{\Delta \varepsilon_{v}^{t+\Delta t}} & =\{\frac{E(\varepsilon )}{1+v}+E(\varepsilon )\lambda \\ & +\frac{\theta (4G+3K)[1-\exp(-\frac{\Delta t}{\theta })]}{3\Delta t}\} \end{array}$$
(25)$$\begin{array}{ll} J=\frac{\Delta \sigma_{ii}^{t+\Delta t}}{\Delta \varepsilon_{jj}^{t+\Delta t}} & =\{E\left(\varepsilon \right)\lambda \\ & +\frac{\theta (3K-2G)[1-\exp(-\frac{\Delta t}{\theta })]}{3\Delta t}\} \end{array}$$
(26)$$\begin{array}{ll} \Phi = & \frac{\Delta \sigma_{ij}^{t+\Delta t}}{\Delta \varepsilon_{v}^{t+\Delta t}}=\{\frac{E(\varepsilon )}{1+v}\\ & +\frac{2G\theta [1-\exp(-\frac{\Delta t}{\theta })]}{\Delta t}\} \end{array}$$

The tangent modulus matrix of the nonlinear viscoelastic constitutive model with damage can be obtained by expanding Equation (23):(27)$$C_{ijkl}^{t+\Delta t}=\left[1-qD^{t+\Delta t}(\varepsilon )\right]\left[\begin{array}{cccccc} \Gamma & J & J & 0 & 0 & 0\\ J & \Gamma & J & 0 & 0 & 0\\ J & J & \Gamma & 0 & 0 & 0\\ 0 & 0 & 0 & \Phi & 0 & 0\\ 0 & 0 & 0 & 0 & \Phi & 0\\ 0 & 0 & 0 & 0 & 0 & \Phi \end{array}\right]$$

#### 2.2.4. Constitutive Modal Verification and Applications

The experimental data validate the constitutive model with damage. Using the three-dimensional damage constitutive model derived in Section 3.3, ABAQUS UMAT subroutines were developed to predict the stress-strain curves of uniaxial tensile specimens at a constant strain of 3%. The reference point for comparison with the experimental curve was chosen as the center point of the middle cross-section of the specimen, as shown in Figure 3. The prediction results show good agreement with the experimental data, which support that the constitutive model can accurately capture the stress-strain response of the propellant.

### 2.3. Finite Element Modeling

To predict the stress and strain response of this structure in detail, a 3D solid model was selected. To simplify the problem without compromising accuracy, a 90° segment model with axis-symmetric boundary conditions on the cut faces was used, owing to the geometry and loading symmetry. A finite element mesh with 17,808 eight-node solid elements (C3D8R) and 22,933 nodes was built for stress and strain analyses, respectively, to acquire their corresponding convergent results, as shown in Figure 4.

To accurately assess the influence of mechanical heterogeneity on HTPB propellant grain structural integrity, the physical parameters of the case, insulation, and two types of propellant were assigned identical values, as displayed in Table 3.

The WLF equation is a method to parameterize the temperature dependence of a quantity, which can be expressed as:(28)$$\mathrm{lg}\alpha_{T}=\frac{-C_{1}(T-T_{r})}{C_{2}+(T-T_{r})}$$
where $\alpha_{T}$ is the time-temperature shift factors; $C_{1}$ and $C_{2}$ are material constants, and two types of propellant are given the same empirical values: $C_{1}$ = 9.151, $C_{2}$ = 173.594; T is the current moment temperature, and $T_{r}$ is the reference temperature.

The model was created with the assumptions and boundary conditions as follows: (i) The thickness of the case was constant. (ii) The insulation liner was elastic. (iii) The outer surface of the case was fixed, and the symmetry plane was set with symmetry constraints. 

For a case-bonded SRM with HTPB propellant, the reference temperature *T_R_* for a strain-free and stress-free condition should be taken to be 8 °C above the propellant cure temperature *T_C_*. The solid rocket motor will be placed at storage temperature *T_S_* after thermal cool down to equilibrium. The calculation conditions are as follows: Step 1: The thermal load step from *T_C_* = 50 °C to storage temperature *T_S_* = 20 °C; Step 2: The ignition pressurization increases up to 10 MPa linearly with a rise time of 0.5 s.

## 3. Results and Discussion

### 3.1. Curing Cooling Load Analysis

The initial elastic modulus at zero strain was used to calculate the mechanical response of two propellant grains under thermal load Step 1. Figure 5 and Figure 6 illustrate the equivalent stress $\sigma_{V}$ and equivalent strain $\varepsilon_{V}$ of the two propellant grains, revealing that their stress and strain responses exhibit spatially varying properties with higher stress and strain areas at the inner bore-free surface of the grain. The maximum equivalent stresses were 0.262 MPa and 0.314 MPa for the two propellant grains, respectively, while their maximum equivalent strains were 0.0653 and 0.0652, respectively.

### 3.2. Mechanical Properties of Spatial Distribution

The propellant exhibits poor load resistance, and prolonged exposure to strain can alter its microstructure, ultimately resulting in changed mechanical properties. According to theory, heterogeneous curing strain can cause spatially varying mechanical properties within the propellant grain. A UMAT subroutine was used to program a mechanical constitutive model with curing strain for two types of propellants. We defined properties at each integration point as a function of curing strain to obtain the spatial variations in propellant mechanical properties. The $E_{e}$, $E_{d}$, $\varepsilon_{th}$, and $\varepsilon_{m}$ of the two types of propellant grains were chosen to further examine the distribution of mechanical properties, as illustrated in Figure 7, Figure 8, Figure 9 and Figure 10.

The constant strain has little impact on the modulus of type I propellant, resulting in essentially identical $E_{e}$ and $E_{d}$ values for the type I propellant grain, as shown in Figure 7. The constant strain can decrease the $E_{e}$ and $E_{d}$ of the type II propellant, causing spatial variation in $E_{e}$ (ranging from 4.336 MPa to 5.851 MPa) and $E_{d}$ (ranging from 1.024 MPa to 1.440 MPa) for the type II propellant grain. Both values are negatively linked to the distribution of curing strain, as shown in Figure 8. The constant strain can also reduce the $\varepsilon_{th}$ of both propellant types, with the type I propellant grain $\varepsilon_{th}$ ranging from 0.053 to 0.058 and the type II propellant grain $\varepsilon_{th}$ ranging from 0.068 to 0.088. Both are inversely related to the distribution of curing strain, as shown in Figure 9. The strain effect can increase the $\varepsilon_{m}$ of the type I propellant but decrease the $\varepsilon_{m}$ of the type II propellant. Therefore, the long-term effect of curing strain may increase the $\varepsilon_{m}$ of type I propellant grain (distributed between 0.555 and 0.747 in Figure 10) while reducing the $\varepsilon_{m}$ of type II propellant grain (distributed between 0.472 and 0.579 in Figure 10).

These results demonstrate that the mechanical properties of propellant within grain exhibit spatial variability, and both types of propellant grain display the characteristic of mechanical heterogeneity when considering the effect of curing strain. Therefore, it is necessary to fully consider the mechanical heterogeneity of the propellant grain for structural integrity assessment.

### 3.3. Mechanical Response

To ensure the safety and reliability of operation, it is critical to assess the structural integrity of the propellant grain during ignition when it is subjected to critical loads. The mechanical responses of the grain under the pressure load of Step 2 are presented in Figure 11 and Figure 12, which clearly show that the inner bore-free surface of the propellant grain is the critical location under ignition pressurization. Comparing the consideration and non-consideration of mechanical heterogeneity of the propellant, the maximum equivalent stress $\sigma_{Vm}$ of type I propellant grain were 0.671 MPa and 0.694 MPa, respectively, with a decrease of approximately 2.8%; and the maximum equivalent strain $\varepsilon_{Vm}$ increased from 0.198 to 0.202, with an increase of approximately 2%.

The curing strain exhibits little effect on the propellant’s elastic modulus in the linear elastic stage and damage stage. However, it extended the linear elastic stage of the stress-strain curve while causing the damage stage of the curve to recede, thereby decreasing the propellant’s modulus in a disguised form when the strain response exceeds the $\varepsilon_{th}$ of the propellant, as shown in Figure 2. Therefore, as the strain response in the grain increases, the decreased disguised modulus leads to the decrease in $\sigma_{V}$ and increase in $\varepsilon_{V}$.

Figure 13 and Figure 14 present the mechanical response of type II propellant grain under the pressure load of Step 2, showing significant differences between considering and not considering the mechanical heterogeneity of the propellant. Considering the mechanical heterogeneity, the maximum equivalent stress $\sigma_{Vm}$ of type II propellant grain decreased from 0.764 MPa to 0.671 MPa, with a decrease of approximately 11.8%. Additionally, the maximum equivalent strain $\varepsilon_{Vm}$ increased from 0.215 to 0.241, with an increase of approximately 12%. This result was due to the curing strains that decreased the modulus of the type II propellant.

For further study on the effect of mechanical heterogeneity on the mechanical response of the grain under ignition pressurization, three paths with obvious gradient changes in mechanical properties were selected for analysis, as shown in Figure 15.

Figure 16 presents the $\sigma_{V}$ of the two propellant grains along paths 1–3. It appears that there are no obvious differences between considering and not considering the mechanical heterogeneity of the propellant on paths 2 and 3, but significant differences on path 1. The reason is that the constant strain exhibits little effect on its modulus in the linear elastic stage for the type I propellant grain, and the mechanical responses on paths 2 and 3 (mostly, $\sigma_{V}$ is below 0.4 MPa, and the corresponding $\varepsilon_{V}$ is below the $\varepsilon_{th}$) are in the linear elastic stage. For the type II propellant grain, small curing strain on path 2 and path 3 causes a slight alteration in mechanical properties. These are the reasons for the small differences between considering and not considering mechanical heterogeneity on path 2 and path 3 of the two propellant grains. The difference on path 1 is primarily concentrated in the middle of the two propellant grains owing to a higher curing strain at this position and relatively apparent mechanical heterogeneity.

Figure 17 shows the $\varepsilon_{V}$ of the two propellant grains along paths 1–3. The differences between path 2 and path 3 are not obvious, and the difference in path 1 is also primarily concentrated in the middle of the two grains.

In conclusion, the mechanical heterogeneity of propellant significantly impacts the mechanical response of the propellant grain under ignition pressurization. The non-uniform distribution of modulus and damage strain threshold is the key parameter that affects the grain’s mechanical response. The increase in modulus results in an increase in the stress response and a decrease in the strain response. When the strain response exceeds $\varepsilon_{th}$, an increase in $\varepsilon_{th}$ improves the propellant’s modulus of the linear elastic stage in a disguised form, which also leads to an increase in the stress response and a decrease in the strain response. Therefore, it is essential to consider the mechanical heterogeneity of the propellant in analyzing the mechanical response of the propellant grain.

### 3.4. Structural Integrity Assessment

To assess the effect of mechanical heterogeneity on the structural integrity of the grain, the damage coefficient $\omega =\varepsilon_{V}/\varepsilon_{th}$ was defined. The propellant appears damaged when ω ≥ 1, but no damage occurs when ω < 1. Figure 18 illustrates the value of ω for two grains along paths 1–3. It is evident that mechanical heterogeneity can significantly increase the ω at the inner bore-free surface of the propellant grain but has little effect on the other parts.

$\eta_{\omega }$ is defined as the damage coefficient ratio between considering and not considering the mechanical heterogeneity of the propellant. Figure 19 shows $\eta_{\omega }$ values on different paths of two grains. It can be observed that, except at the root of the artificial debonding layer, $\eta_{\omega }$ values are slightly greater than 1 along path 1 of both propellant grains. On the inner bore-free surface, $\eta_{\omega }$ increased by approximately 11% for type I propellant grain and approximately 45% for type II propellant grain. This increase is attributable to two factors: On the one hand, $\varepsilon_{th}$ values decrease with the increasing effect of curing strain for both types of propellant; on the other hand, considering the effect of mechanical heterogeneity, the strain response in the middle of the inner hole of both grains increases.

To assess the structural integrity of the propellant grain, the safety factor SF was introduced and defined as SF = $\varepsilon_{m}/\varepsilon_{V}$. The Von Mises strain criterion was used to analyze the SF of two propellant grains under ignition pressurization. Figure 20 and Figure 21 show the safety factor contours of the two propellant grains. Compared to not considering mechanical heterogeneity, considering it led to an increase in the SF of type I propellant grain from 2.771 to 3.669, representing an increase of approximately 32.41%; the SF of type II propellant grain decreased from 1.933 to 2.662, a reduction of approximately 27.4%.

The safety factor of the propellant grain is directly affected by $\varepsilon_{V}$ of the grain and the $\varepsilon_{m}$ of the propellant. The factor analysis method was used to calculate the degree of influence of these two factors, as shown in Table 4. When considering mechanical heterogeneity, the SF of type I propellant grain increased from 2.771 to 3.669, representing a 29.34% increase. The contribution of $\varepsilon_{V}$ and $\varepsilon_{m}$ to the increase in SF was −6.65% and 106.65%, respectively; the SF of type II propellant grain decreased from 2.662 to 1.933, a reduction of 27.38%. The contribution of $\varepsilon_{V}$ and $\varepsilon_{m}$ to the decrease in $\varepsilon_{V}$ was 37.04% and 62.96%, respectively. It can be concluded that the change in SF for both propellant grains is primarily influenced by propellant $\varepsilon_{m}$.

## 4. Conclusions

The study investigates the effect of curing strain-induced mechanical heterogeneity in an HTPB propellant on the grain structural integrity during ignition pressurization. The conclusions are as follows:(1)When considering the influence of load on propellant mechanical properties, the spatial variability of propellant grain mechanical properties occurs. The range of variation is related to the load size and its effect on propellant mechanical properties.(2)The non-uniform distribution of modulus and damage strain threshold are key parameters affecting the grain’s mechanical response. Increasing the modulus leads to an increased stress response and decreased strain response. A disguised increase in the linear elastic stage modulus results from the $\varepsilon_{th}$ increase when the strain response exceeds $\varepsilon_{th}$, which also increases the stress response and decreases the strain response.(3)The safety factor based on the equivalent strain failure criterion of the grain is directly affected by the $\varepsilon_{V}$ of the grain and the $\varepsilon_{m}$ of the propellant. The change in SF of the two propellants is primarily affected by the $\varepsilon_{m}$ of the propellant.

## Figures and Tables

**Figure 1 materials-16-04590-f001:**
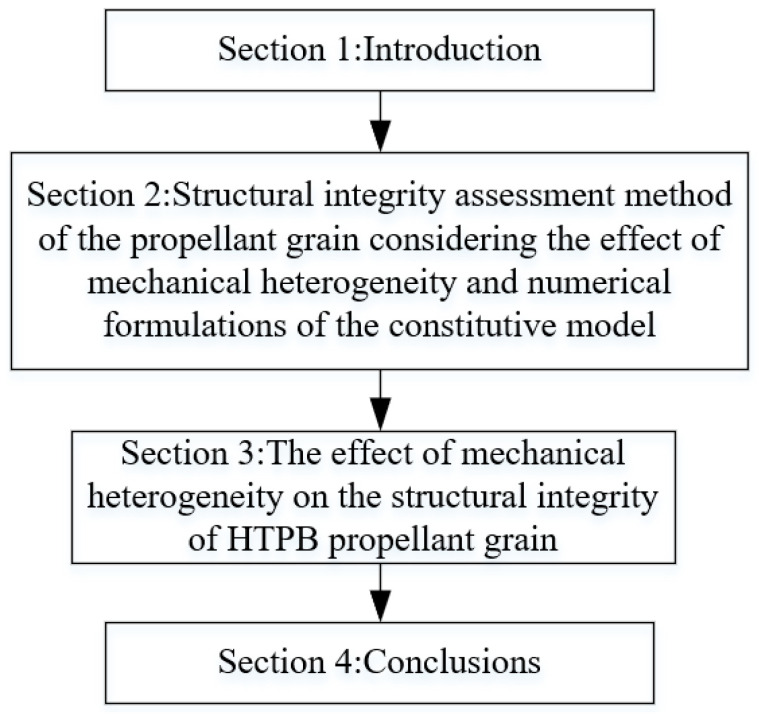
The organization of the paper.

**Figure 2 materials-16-04590-f002:**
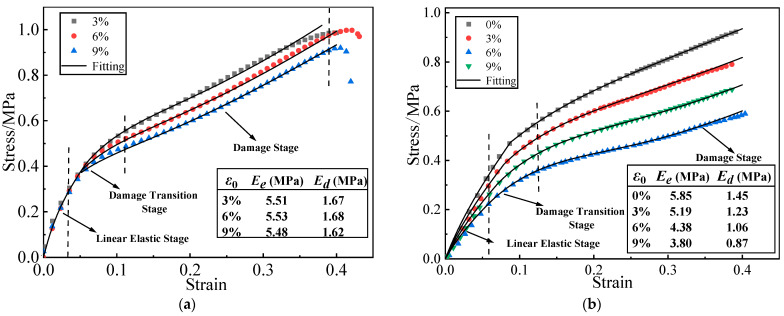
Two typical stress-strain curves of HTPB propellant: (**a**) Type I HTPB propellant stress-strain curve; (**b**) Type II HTPB propellant stress-strain curve.

**Figure 3 materials-16-04590-f003:**
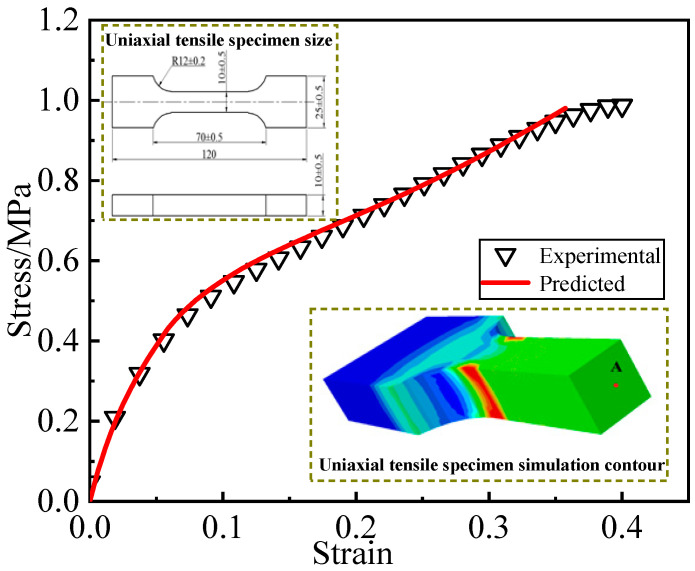
Comparison of experimental curves with numerical prediction results.

**Figure 4 materials-16-04590-f004:**
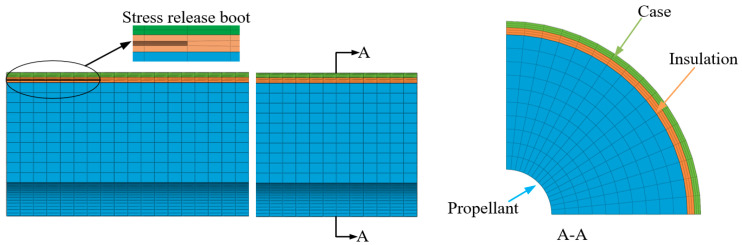
Mesh division of solid rocket motors.

**Figure 5 materials-16-04590-f005:**
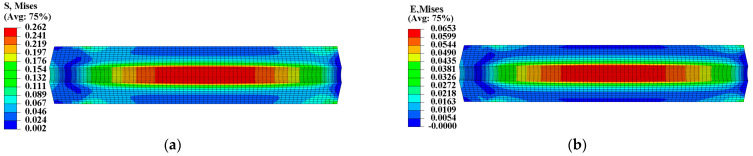
Mechanical response of type I propellant grain: (**a**) Curing stress contour; (**b**) curing strain contour.

**Figure 6 materials-16-04590-f006:**
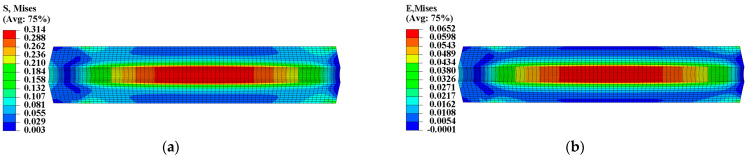
Mechanical response of type II propellant grain: (**a**) Curing stress contour; (**b**) curing strain contour.

**Figure 7 materials-16-04590-f007:**
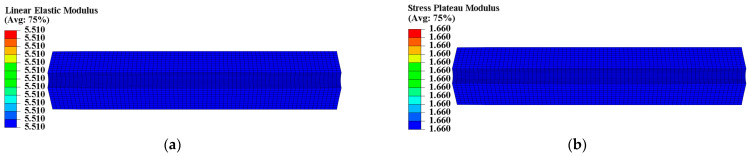
Modulus distribution of type I propellant grain: (**a**) Distribution of $E_{e}$; (**b**) distribution of $E_{d}$.

**Figure 8 materials-16-04590-f008:**
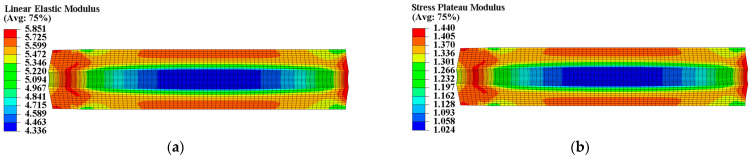
Modulus distribution of type II propellant grain: (**a**) Distribution of $E_{e}$; (**b**) distribution of $E_{d}$.

**Figure 9 materials-16-04590-f009:**
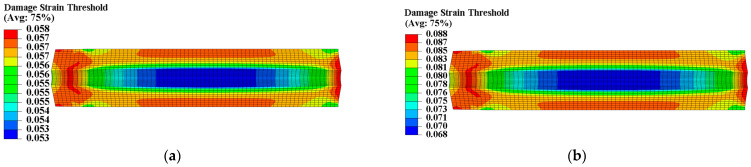
Damage strain threshold distribution of propellant grain: (**a**) Distribution of $\varepsilon_{th}$ in type I grain; (**b**) distribution of $\varepsilon_{th}$ in type II grain.

**Figure 10 materials-16-04590-f010:**
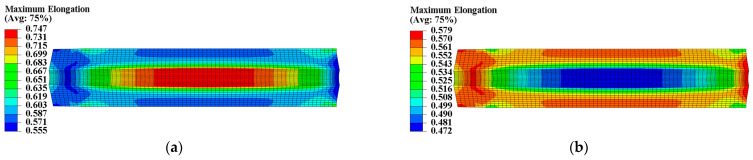
Maximum elongation distribution of propellant grain: (**a**) Distribution of $\varepsilon_{m}$ in type I grain; (**b**) distribution of $\varepsilon_{m}$ in type II grain.

**Figure 11 materials-16-04590-f011:**
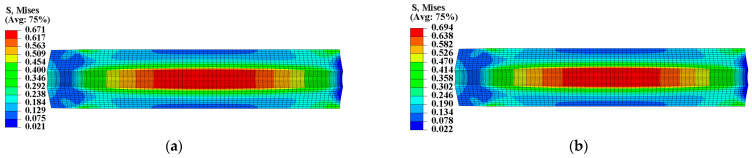
Equivalent stress of type I propellant grain: (**a**) Considering the mechanical heterogeneity; (**b**) not considering the mechanical heterogeneity.

**Figure 12 materials-16-04590-f012:**
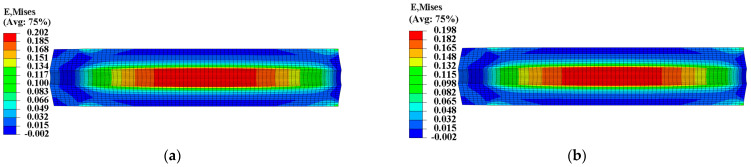
Equivalent strain of type I propellant grain: (**a**) Considering the mechanical heterogeneity; (**b**) not considering the mechanical heterogeneity.

**Figure 13 materials-16-04590-f013:**
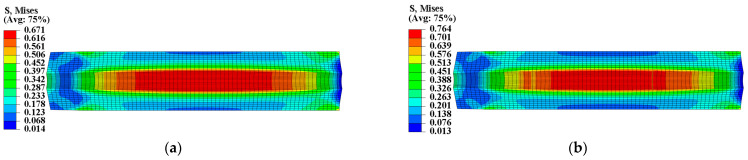
Equivalent stress of type II propellant grain: (**a**) Considering the mechanical heterogeneity; (**b**) not considering the mechanical heterogeneity.

**Figure 14 materials-16-04590-f014:**
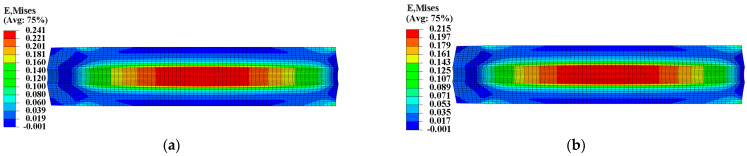
Equivalent strain of type II propellant grain: (**a**) Considering the mechanical heterogeneity; (**b**) not considering the mechanical heterogeneity.

**Figure 15 materials-16-04590-f015:**

Path selection.

**Figure 16 materials-16-04590-f016:**
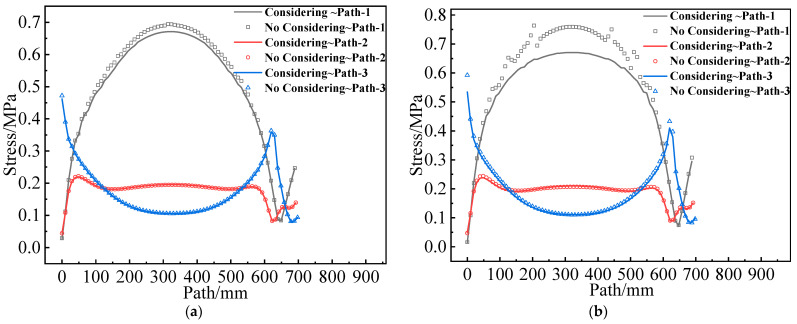
Equivalent stress varies along paths 1–3: (**a**) Type I propellant grain; (**b**) Type II propellant grain.

**Figure 17 materials-16-04590-f017:**
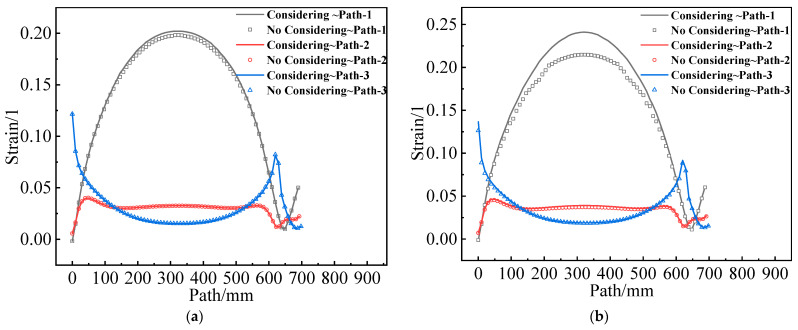
Equivalent strain varies along paths 1–3: (**a**) Type I propellant grain; (**b**) Type II propellant grain.

**Figure 18 materials-16-04590-f018:**
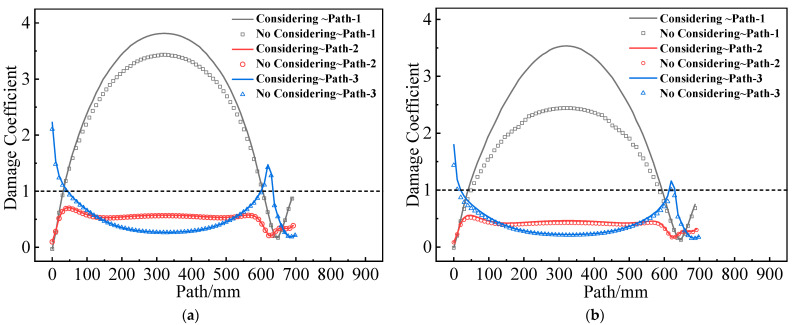
Damage coefficient varies along paths 1–3: (**a**) Type I propellant grain; (**b**) Type II propellant grain.

**Figure 19 materials-16-04590-f019:**
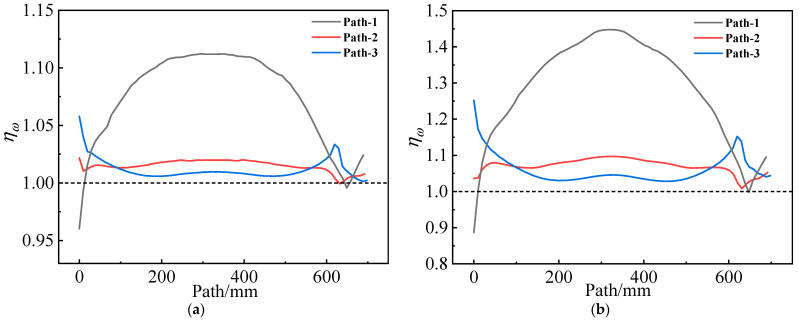
Damage coefficient ratio varies along paths 1–3: (**a**) Type I propellant grain; (**b**) Type II propellant grain.

**Figure 20 materials-16-04590-f020:**
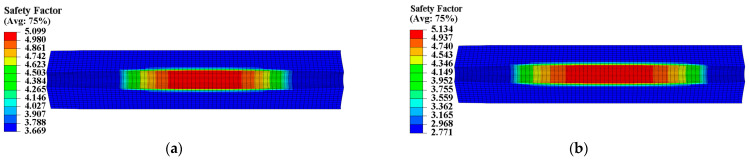
Safety factor contour of type I propellant grain: (**a**) Considering the mechanical heterogeneity; (**b**) not considering the mechanical heterogeneity.

**Figure 21 materials-16-04590-f021:**
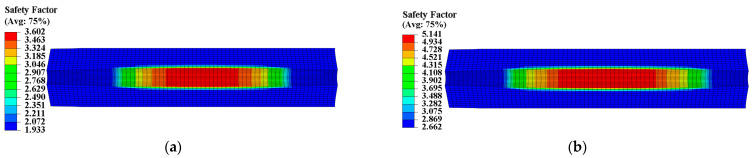
Safety factor contour of type II propellant grain: (**a**) Considering the mechanical heterogeneity; (**b**) not considering the mechanical heterogeneity.

**Table 1 materials-16-04590-t001:** Parameters of propellant constitutive model.

Parameters	Type I Propellant	Type II Propellant
$\sigma_{m}$/MPa	22.18	−82.33 × $\varepsilon_{0}$ + 20.23
$E_{1}$/MPa	8.91	−11 × $\varepsilon_{0}$ + 3.57
θ/s	0.79	1.3
α/1	0.20	0.21
q/1	0.27 × exp(2.27 × $\varepsilon_{0}$ + 0.44)	−1.13 × $\varepsilon_{0}$ + 0.57
η/1	0.13–0.33 × $\varepsilon_{0}$	−0.35 × $\varepsilon_{0}$ + 0.17
m/1	0.98	6.90 × $\varepsilon_{0}$ + 0.91
$\varepsilon_{th}$/1	0.06–0.07 × $\varepsilon_{0}$	−0.31 × $\varepsilon_{0}$ + 0.09

Note: The constitutive model parameters of type I propellant are provided in Ref. [26]; the constitutive model parameters of type II propellant are determined by fitting the curve in Figure 2b.

**Table 2 materials-16-04590-t002:** Variation of maximum elongation $\varepsilon_{m}$ of propellant with curing strain.

	Type I Propellant	Type II Propellant
$\varepsilon_{m}$/1	2.94 × $\varepsilon_{0}$ + 0.56	−1.63 × $\varepsilon_{0}$ + 0.58

**Table 3 materials-16-04590-t003:** Material parameters of solid rocket motor parts.

	Case	Insulation	HTPB Propellant
E/MPa	1.96 × 10^5^	30	/
ρ/(kg/m^3^)	7850	1220	1735
v/1	0.28	0.498	0.498
α/(1/K)	1.1 × 10^−5^	8.6 × 10^−5^	8.6 × 10^−5^

**Table 4 materials-16-04590-t004:** Analysis of influence factors on safety factor of two propellant grains.

		Homogeneous Material	Inhomogeneous Material	Variation	Influence Degree
Type I propellant	$\varepsilon_{V}$/1	0.198	0.202	2.02%	−6.65%
	$\varepsilon_{m}$/1	0.555	0.747	34.60%	106.65%
	SF/1	2.771	3.669	32.41%	100%
Type II propellant	$\varepsilon_{V}$/1	0.215	0.241	12.09%	37.04%
	$\varepsilon_{m}$/1	0.578	0.472	−18.34%	62.96%
	SF/1	2.662	1.933	−27.38%	100%

## Data Availability

The data that support the findings of this study are available on request from the corresponding author.

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
