# Peer review of "Effect of Mechanical Heterogeneity on the Structural Integrity of HTPB Propellant Grain"

_materials, 2023, doi:10.3390/ma16134590_

Round 1

Reviewer 1 Report

The paper deals with an interesting approach from intellectual point of view. However, there is no novelty in the work and no substantial contribution os observed. 

Data is explained but no fuerther analysis was performed. I recommend the rejection of this work.

Reviewer 2 Report

line no 16...not clear..increase in stress response means? decrease in strain response means??

line 66-67...incorrect grammatically

line 136...something wrong...results and discussion??

line 149...check spellings

line 225 check spellings

table 3- check modulus value

boundary condition pics/sketches? Try to show clearly

If the properties are varying in different directions, then the results also should be shown in different directions ex, ey, ez strains, then why mises strain?

Model is 3D, right? then why all results views look 2D only??

Try to show 3D model results alongwith mesh and contour

OK

Minor spellcheck editing required

Reviewer 3 Report

The reviewed article concerns a study of the effect of mechanical heterogeneity on the structural integrity of polybutadiene fuel granules with hydroxyl ends (HTPB). The authors of this work proposed a gradient finite element method for direct assessment of structural integrity, where inhomogeneous mechanical properties of rocket fuel grains were constructed and evaluated. The authors have done a lot of work. The writing is well written and easy to follow. The figures are well structured and clear. Well done. However, there are some issues that I consider necessary to discuss before accepting this work in Materials.

1.       The annotation should be improved and some numerical values should be added so that it reflects the content well. This is due to the novelty and importance of the current study compared to previous recent works found in the open literature, should be well explained.

2.       The references part should be updated and some recent works in the field should be cited. (https://doi.org/10.1016/j.saa.2022.121869, https://doi.org/10.3390/ma13133031, https://doi.org/10.1038/s41598-022-22726-8)

3.       The problem lies in the basic premise of comparing the calculated data with the parameters obtained by the microscope. Why didn't you check it? So the basic concept is not really understood. And the result is this: it doesn't work.

4.       It will be helpful if the authors could add the Stress nephograms and SEM micrographs of the investigated compounds.

5.       “The paper is organized as follows: Section 2 introduces the structural integrity assessment method of the propellant grain considering the effect of mechanical heterogeneity. Section 3 establishes numerical formulations of the constitutive model. Section 4 analyzes the effect of mechanical heterogeneity on the structural integrity of HTPB propellant grain. Finally, Section 5 summarizes the conclusions.” This introductory insert needs to be transformed for a better perception of the work, make it in graphic form.

Round 2

Reviewer 3 Report

Thanks for the corrections to the manuscript.

Author Response

Thank you very much for your continued interest in the manuscript and your pertinent comments.